# Proteomic associations with cognitive variability as measured by the Wisconsin Card Sorting Test in a healthy Thai population: A machine learning approach

Chen Chen[1]*, Bupachad Khanthiyong[2], Benjamard Thaweetee-Sukjai[3], Sawanya Charoenlappanit[4], Sittiruk Roytrakul[4], Phrutthinun Surit[5], Ittipon Phoungpetchara[6,7], Samur Thanoi[8], Gavin P. Reynolds[9,7], Sutisa Nudmamud-Thanoi[6,7]*

1 Faculty of Medical Science, Medical Science graduate program, Naresuan University, Phitsanulok, Thailand, 2 Faculty of Medicine, Bangkokthonburi University, Bangkok, Thailand, 3 School of Medicine, Mae Fah Luang University, Chiang Rai, Thailand, 4 National Centre for Genetic Engineering and Biotechnology, National Science and Technology Development Agency, Pathum Thani, Thailand, 5 Department of Biochemistry, Faculty of Medical Science, Naresuan University, Phitsanulok, Thailand, 6 Department of Anatomy, Faculty of Medical Science, Naresuan University, Phitsanulok, Thailand, 7 Centre of Excellence in Medical Biotechnology, Faculty of Medical Science, Naresuan University, Phitsanulok, Thailand, 8 School of Medical Sciences, University of Phayao, Phayao, Thailand, 9 Biomolecular Sciences Research Centre, Sheffield Hallam University, Sheffield, United Kingdom

* wzhnky@hotmail.com (CC); sutisat@nu.ac.th (SN-T)

## Abstract

Inter-individual cognitive variability, influenced by genetic and environmental factors, is crucial for understanding typical cognition and identifying early cognitive disorders. This study investigated the association between serum protein expression profiles and cognitive variability in a healthy Thai population using machine learning algorithms. We included 199 subjects, aged 20 to 70, and measured cognitive performance with the Wisconsin Card Sorting Test. Differentially expressed proteins (DEPs) were identified using label-free proteomics and analyzed with the Linear Model for Microarray Data. We discovered 213 DEPs between lower and higher cognition groups, with 155 upregulated in the lower cognition group and enriched in the IL-17 signaling pathway. Subsequent bioinformatic analysis linked these DEPs to neuroinflammation-related cognitive impairment. A random forest model classified cognitive ability groups with an accuracy of 81.5%, sensitivity of 65%, specificity of 85.9%, and an AUC of 0.79. By targeting a specific Thai cohort, this research provides novel insights into the link between neuroinflammation and cognitive performance, advancing our understanding of cognitive variability, highlighting the role of biological markers in cognitive function, and contributing to developing more accurate machine learning models for diverse populations.

## Introduction

Cognitive function denotes the higher-order mental processes in the brain that gather and process information, reflecting brain activity. In adults, inter-individual cognitive variability is pervasive [1], influenced by both genetic and environmental factors [2,3], including

**Data availability statement:** The author-generated code and the datasets generated and/or analyzed during the current study are uploaded as supplementary files.

**Funding:** S.N-T. received partial support from the Global and Frontier Research University Fund, Naresuan University (grant number R2567C003; https://www.nu.ac.th/) and the Reinventing University Program 2024, Ministry of Higher Education, Science, Research and Innovation (MHESI), Thailand (grant number R2567A141; https://www.mhesi.go.th/). The funders had no role in study design, data collection and analysis, decision to publish, or preparation of the manuscript. There was no additional external funding received for this study.

**Competing interests:** The authors declare that they have no known competing financial interests or personal relationships that could have appeared to influence the work reported in this paper.

gender, age [4], and lifestyle choices [3]. For example, variability in cognitive trajectories has been observed in community-dwelling older adults across different cognitive domains, such as episodic memory, vocabulary, executive function, attention, and psychomotor speed [5]. Differences in cognitive responses to the same physical exercise among individuals are also apparent [6,7]. Studying normal cognitive variability is essential for establishing a baseline understanding of typical cognition, which is crucial for identifying anomalies that may signal early cognitive disorders [8]. Additionally, understanding normal cognitive differences aids in developing strategies and educational practices that ensure the society is tailored to support diverse cognitive strengths [9].

Genetic and environmental factors may exhibit complex direct and indirect interactions that influence normal cognitive variability. Genetic factors contribute approximately 50% to 70% of cognitive variability at the population level. However, genetic influences on cognitive function increase from birth to maturity, with these effects being pronounced in more advantaged socioeconomic groups [10]. Previous research indicates that differences in cognitive flexibility between healthy Thai males and females fluctuate across age, with such cognitive sex differences notable in subjects over 60 years [11]. Additionally, experiences and knowledge gained through education alter activity in cholinergic pathways, leading to an attenuation of these sex-dependent cognitive differences [12]. This complexity poses a significant challenge for researchers attempting to unravel the contributing elements and their interactions that influence normal cognitive variability. Thus, using information technology and computer-based algorithms to probe these various complex interconnections offers great potential for enhancing our understanding of cognitive variability.

Machine learning (ML) algorithms serve as robust approaches for processing large-scale datasets and detecting intricate patterns that conventional statistical methods might fail to reveal [13,14]. ML models have been employed to classify cognitive profiles [15–17] and to predict cognitive health in the global population [18–20]. When using ML algorithms, the racial and ethnic background of subjects is an essential issue to consider [21], since racial bias is a prevalent challenge facing ML in human studies [22], with consequences that can lead to racial disparities in healthcare access and outcomes [23].

Given that cognitive function reflects brain activity, alterations in molecular factors such as neuronal and neurotransmission-related proteins may offer valuable insights into cognitive variability in healthy individuals. Indeed, previous studies have demonstrated that biomarkers related to brain activity can serve as indicators of cognitive function [24–26]. Accordingly, this study aims to investigate the association between serum protein expression profiles and one measure of cognitive variability, as assessed by the Wisconsin Card Sorting Test (WCST), in a healthy Thai population using ML algorithms. By targeting a specific Thai cohort, this research contributes to the development of ML models tailored to diverse populations in cognitive studies.

## Methods

### Participants and data

This study included 199 healthy Thai subjects, ranging in age from 20 to 70 years, with samples collected and cognitive tests conducted from October 20, 2014, to August 25, 2018, as previously reported [27]. The researchers assessed the archived samples and data from July 6, 2022, to July 6, 2023, for the preparation of this publication; all subjects were assigned a study ID that concealed individual identities. The cognitive performance of each participant was evaluated using the WCST, a test that measures cognitive flexibility [28,29]—the capacity to adapt the behavioral response mode to changing conditions—which is commonly employed

to assess frontal lobe function, especially the prefrontal cortex [30]. 3 ml of blood was collected from the cubital vein of each subject immediately following the completion of the WCST. The serum protein expression profiles of the subjects were then analyzed using the label-free proteomics method, as previously described [11].

The mean and standard deviation (SD) of the percentage of total errors (%Errors) in WCST were computed in male and female subjects separately, covarying for age [11]. A negative correlation has been observed between WCST %Errors and the Full-Scale Intelligence Quotient (FSIQ) [31,32], a metric used to assess a person's overall level of general cognitive and intellectual functioning [33]. Male and female subjects with %Errors > 1SD above their sex-specific mean were considered to have poor cognitive performance and were assigned to the lower cognitive ability group [34]. All methods were performed in compliance with relevant guidelines and regulations (Declaration of Helsinki). The Institutional Review Board (IRB) of Naresuan University, Thailand approved this research (COA No. 0262/2022). Participation was voluntary, and written informed consent was obtained from each subject.

## Bioinformatic analysis

Differentially expressed proteins (DEPs) between the two cognition groups were identified using the Linear Model for Microarray Data (LIMMA) approach in R version 4.2.3. The LIMMA approach applies linear modeling to the expression data and employs empirical Bayesian techniques to adjust the standard errors of the estimated log-fold changes [35]. The significance of the differential expression was determined using the adjusted P-value, with a threshold set at $P \leq 0.01$ [36]. This statistical rigor enhances the validity of our findings, ensuring that the results are both robust and biologically meaningful.

Pathway analysis and Gene Ontology (GO) enrichment analysis were performed on the website of Annotation, Visualization, and Integrated Discovery (DAVID). DAVID utilizes Fisher's Exact Test to assess the enrichment of GO terms and pathways among the DEPs, ensuring that these results are statistically significant and reliable [37,38]. The protein-protein interaction network of the identified DEPs was studied using Pathway Studio version 12.5 [39]. DEP expression levels were displayed by Multi-Experiment Viewer (MeV) version 4.9.0 [40]. $P \leq 0.05$ was considered significant.

## Preprocessing

The training and testing datasets were respectively proportioned at 0.6 (n = 119) and 0.4 (n = 80) of the total sample to optimize the balance between training and validation sets. This approach is designed to enhance the model's generalization capability and minimize the risk of overfitting. By allocating a sufficient portion of the data to the training set while preserving a substantial validation set, we can achieve more reliable and robust model performance, consistent with best practices in ML [41].

To address the relatively small proportion of subjects defined as having poor cognitive performances, the synthetic minority over-sampling technique (SMOTE) was employed [42]. SMOTE was applied exclusively to the training dataset during cross-validation to minimize overfitting. The testing dataset, excluded from SMOTE or cross-validation procedures, was used to evaluate the final performance metrics of the model [18].

The DEPs that were significantly enriched in cognition-related pathways, as identified by GO enrichment analysis, pathway analysis, and protein-protein interaction network analysis, were selected as model variables. Age was also included in the model since it is strongly connected to cognitive impairment in the healthy population [43,44].

### Machine learning model

The ML processes were conducted using the *tidy models* meta-package in R [45]. This approach facilitated the creation of a unitary preprocessing dataset, enabling reliable comparisons across various ML models, including K-nearest neighbors, regression, decision tree, and random forest (RF). The RF model was selected for this study due to its superior performance. The analysis code can be found in S1 File.

The principles and procedures of the RF model are well-documented [46–48]. In brief, the RF is an ensemble learning method based on decision tree algorithms and is applicable to both classification and regression analysis [49]. The significance of each variable in the final model is assessed using a ranked measure of variable importance.

### Model validation

Cross-validation is a technique employed to evaluate the performance and generalizability of an ML model. It involves dividing the data into multiple subsets, training the model on certain subsets, and validating it on others. This method minimizes overfitting and yields a more accurate estimate of model performance [50]. In this study, a 10-fold cross-validation was conducted on the training dataset to evaluate the models' performances.

In addition, model hyperparameters were tuned during the cross-validation [45]. Specifically, the following hyperparameters were tuned in this study: the number of trees was 1000, the minimum number of data points required for a node further splitting was 2, and 10 predictors were randomly sampled at each split when building the tree models.

### Performance metrics

In this study, we utilized several performance metrics to evaluate the RF model comprehensively. Overall accuracy measures the proportion of correctly classified instances among all instances. Matthews correlation coefficient (MCC) is a balanced measure that accounts for true and false positives and negatives, providing a comprehensive understanding of model performance [51]. The $F_1$-score, which is the harmonic mean of precision and recall, offers insight into the balance between the precision and the completeness of the model [52]. Finally, the Area Under the Receiver Operating Characteristic Curve (AUC) assesses the model's ability to distinguish between classes, offering a summary of performance across all classification thresholds [53]. Fig 1 shows the overall working procedure of this study.

## Results

### Demographic data of the study population

The demographic characteristics of the study population are detailed in Table 1 (see full data set in S1 Table). The mean age of the participants was $45.6 \pm 19.3$ years, with 55.3% being female (n = 110). The cutoff point as defined by %Errors > 1SD from the mean in males was 62.3, whereas in females, it was 67.4.

### Bioinformatic analysis

There were 213 differentially expressed proteins identified between the poor and higher cognition groups, with 155 DEPs being upregulated in the poor cognition group and the remaining 58 DEPs being downregulated (see Fig 2). Data from the subsequent GO enrichment analysis demonstrated that DEPs were substantially enriched in the following biological pathways: regulation of protein stability (P = 0.01), macromolecule methylation (P = 0.03), regulation of neuron projection development (P = 0.04), and retinoic acid catabolic process

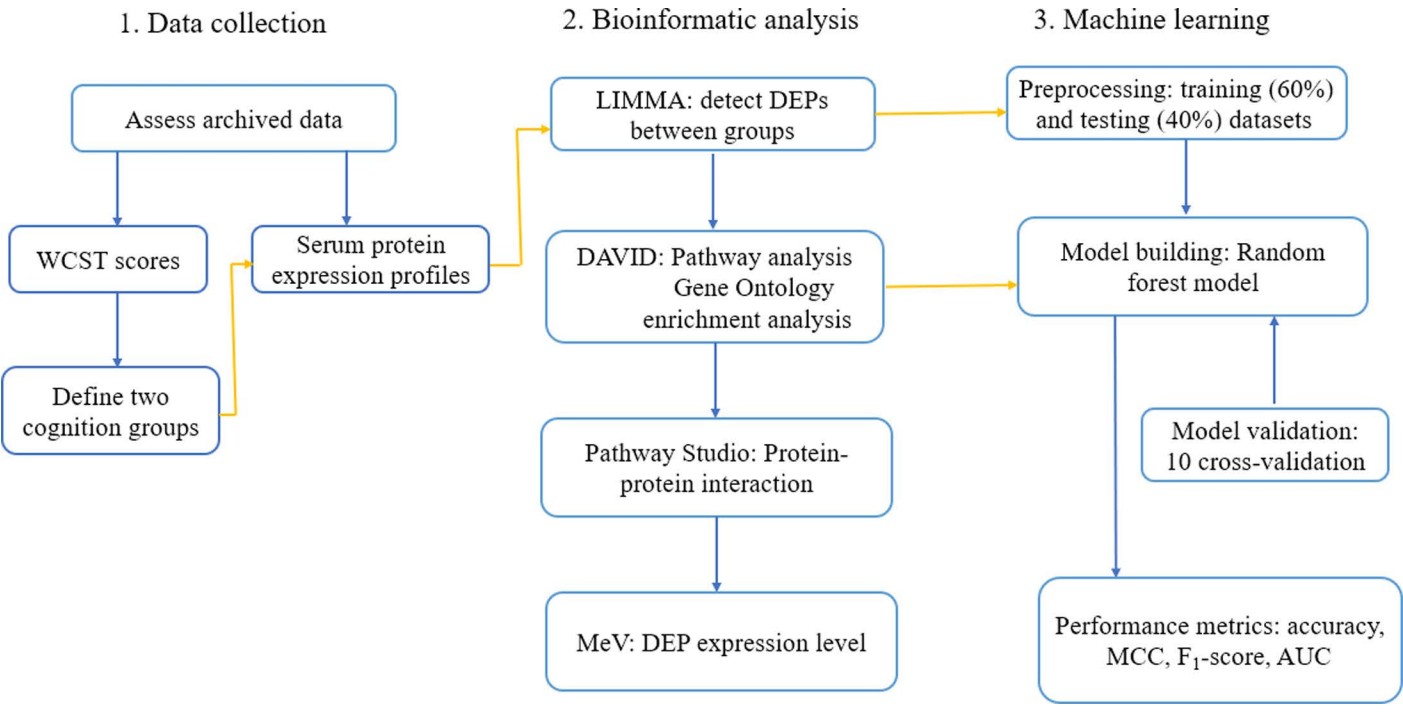

**Fig 1. Overall working procedure of this study.**

**Table 1. Demographic data of the study population.**

| | Lower cognitive ability (n = 41) | Higher cognitive ability (n = 158) |
|---|---|---|
| Age, years | 51.5 ± 17.5 | 44.0 ± 19.6 |
| Sex, female | 22 (53.7%) | 88 (55.7%) |
| Education, secondary & tertiary | 14 (34.1%) | 93 (58.9%) |
| WCST %Errors | 69.6 ± 4.78 | 48.6 ± 10.0 |

Data is shown as mean ± SD

(P = 0.04), as shown in Table 2. Regarding the pathway analysis [54–56], these DEPs showed significant enrichment in the IL-17 signaling pathway (P = 0.05); see Table 3.

Furthermore, protein-protein interaction (PPI) network analysis indicated that most of the 16 DEPs enriched in the aforementioned pathways were linked to neuroinflammation-related cognitive impairment (see full data set in S1 Table). The other four DEPs found to be involved in this PPI network were serotonin receptor 7 (HTR7), metabotropic glutamate receptor 4 (GRM4), choline transporter-like protein 2 (SLC44A2), and pro-adrenomedullin (ADM) (see Fig. 3). Glutamatergic [57], serotonergic [58], and cholinergic [59] systems have been shown to have a role in cognitive impairment. Additionally, ADM has been suggested as a potential biomarker for cognitive impairment [60,61]. As a result, all those 20 proteins were included as variables in the RF model.

## Model performance

The RF model achieved a test classification accuracy of 81.5% (see Table 4). The model's sensitivity (true positive rate) was estimated to be 65%, while the specificity (true negative rate) was 85.9%. The AUC (0.79) indicates good binary classification performance [62] (see Fig. 4).

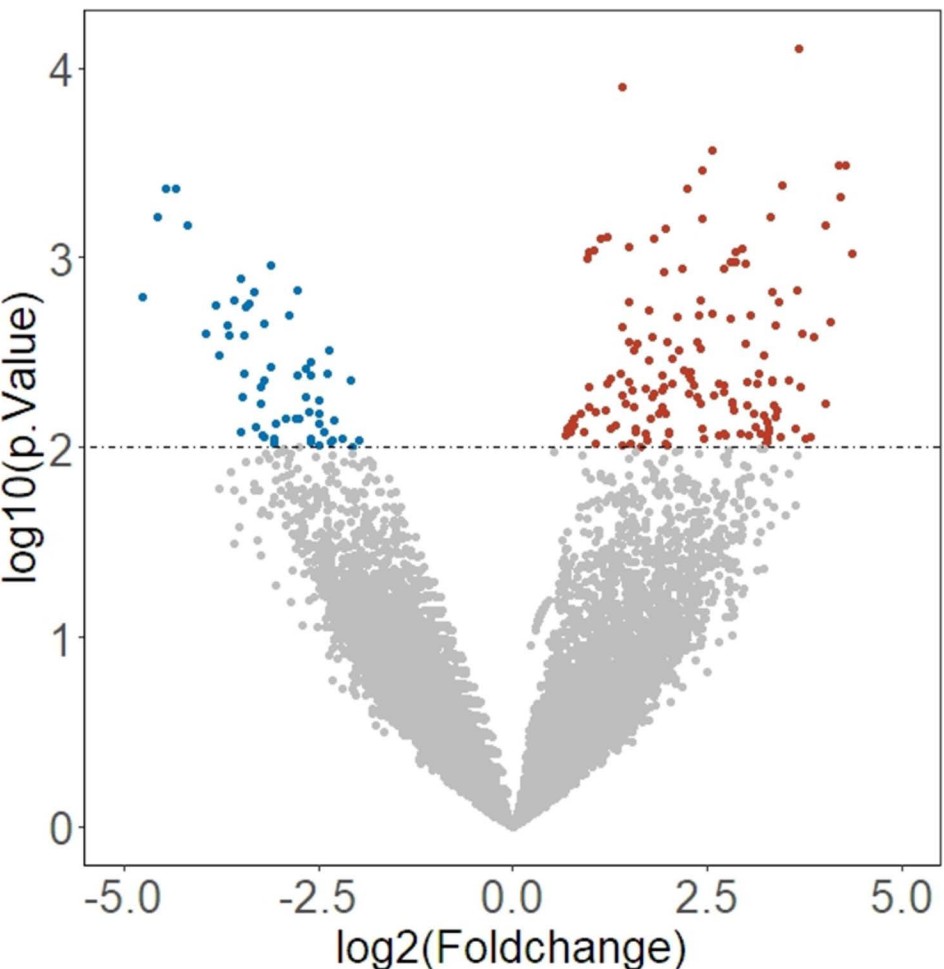

**Fig 2. Volcano plot of DEPs between poor and normal cognition groups.** Color blue represents DEPs that are downregulated in the poor cognition group, red represent DEPs unregulated in poor cognition group.

**Table 2. Gene Ontology analysis of DEPs between lower and higher cognitive ability groups.**

| Number | Pathway ID | Pathway | Mapped DEPs | Fold Enrichment | P-value |
|---|---|---|---|---|---|
| 1 | GO:0031647 | Regulation of protein stability | GET4, CCAR2, KAT2A, UPS25, INSC | 5.84 | 0.01 |
| 2 | GO:0043414 | Macromolecule methylation | FTSJ3, METTL4 | 55.5 | 0.03 |
| 3 | GO:0010975 | Regulation of neuron projection development | NCS1, BRSK2, NTNG1 | 8.99 | 0.04 |
| 4 | GO:0034653 | Retinoic acid catabolic process | CRABP1, CYP2W1 | 44.3 | 0.04 |

**Table 3. Pathway analysis of DEPs between lower and higher cognitive ability groups.**

| Number | Pathway ID | Pathway | Mapped DEPs | Fold Enrichment | P-value |
|---|---|---|---|---|---|
| 1 | hsa04657 | IL-17 signaling pathway | MAPK6, CCL7, MAPK9, UPS25 | 4.56 | 0.05 |

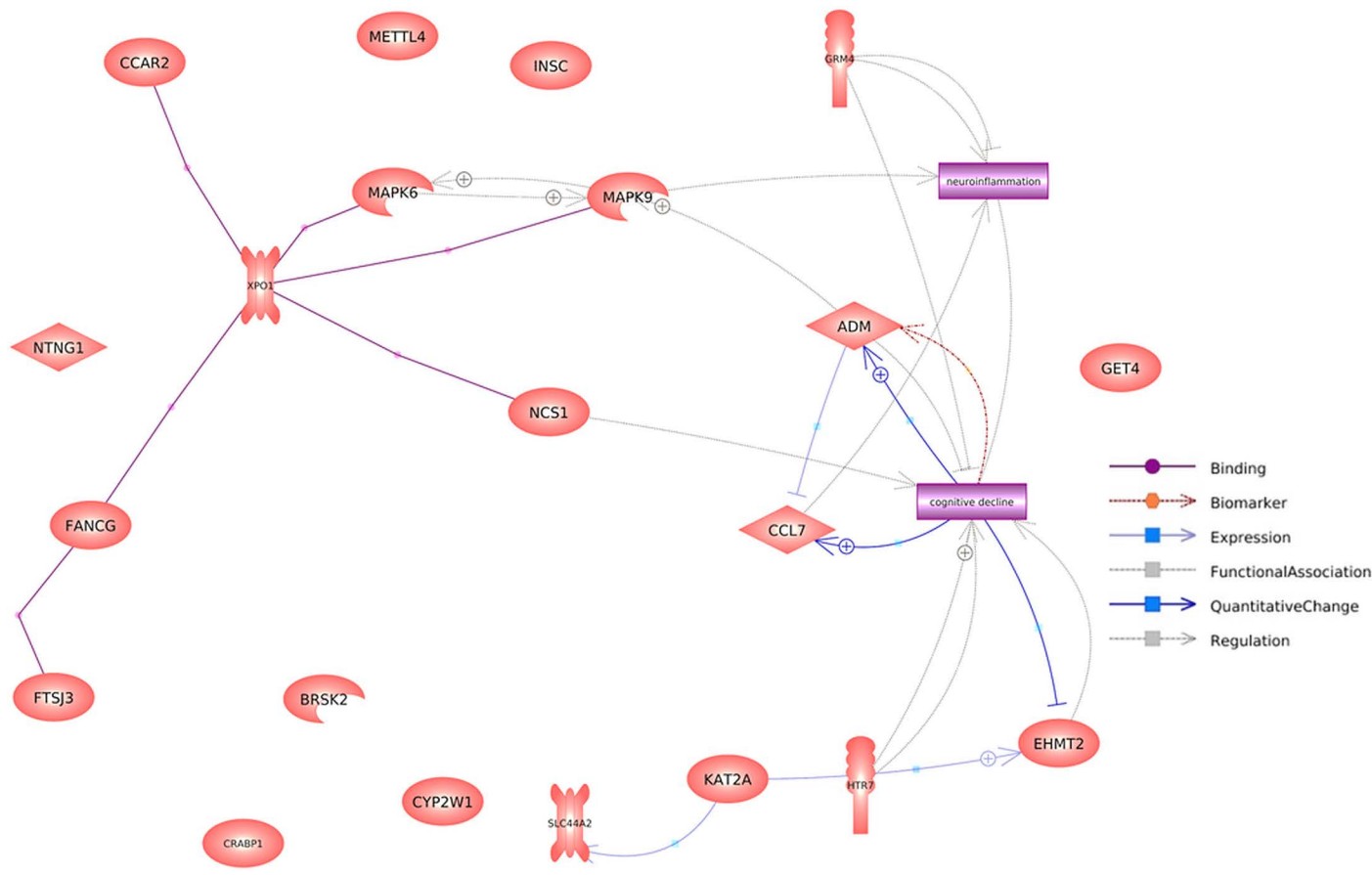

**Fig 3. Protein-protein interaction network of selected DEPs.**

**Table 4. Performance metrics of the RF model.**

| accuracy | sensitivity | specificity | MCC | $F_1$-score | AUC |
|---|---|---|---|---|---|
| 0.815 | 0.65 | 0.859 | 0.478 | 0.595 | 0.790 |

## Importance of the model variables

Fig 5 depicts the 10 most influential variables. MAPK9 (P45984) had the strongest influence on the classification model, and this DEP was significantly enriched in the IL-17 signaling pathway. Age, ADM (P35318), HTR7 (P34969), SLC44A2 (Q8IWA5), and GRM4 (Q14833) are among the most important variables, as expected. The remaining four variables are NTNG1 (Q9y2I2), FTSG3 (Q8IY81), CCAR2 (Q8N163), and UPS25 (Q9UHP3), in descending order of importance. In the poor cognition group, 7 of those 9 DEPs were upregulated, whereas SLC44A2 and GRM4 were downregulated (see Fig. 6).

## Discussion

This study built a classification model using ML algorithms to detect healthy Thai subjects with poor cognitive performance based on proteomic data. The RF model showed good performance with an accuracy of 0.815, a specificity of 0.859, and an AUC of 0.790, despite a

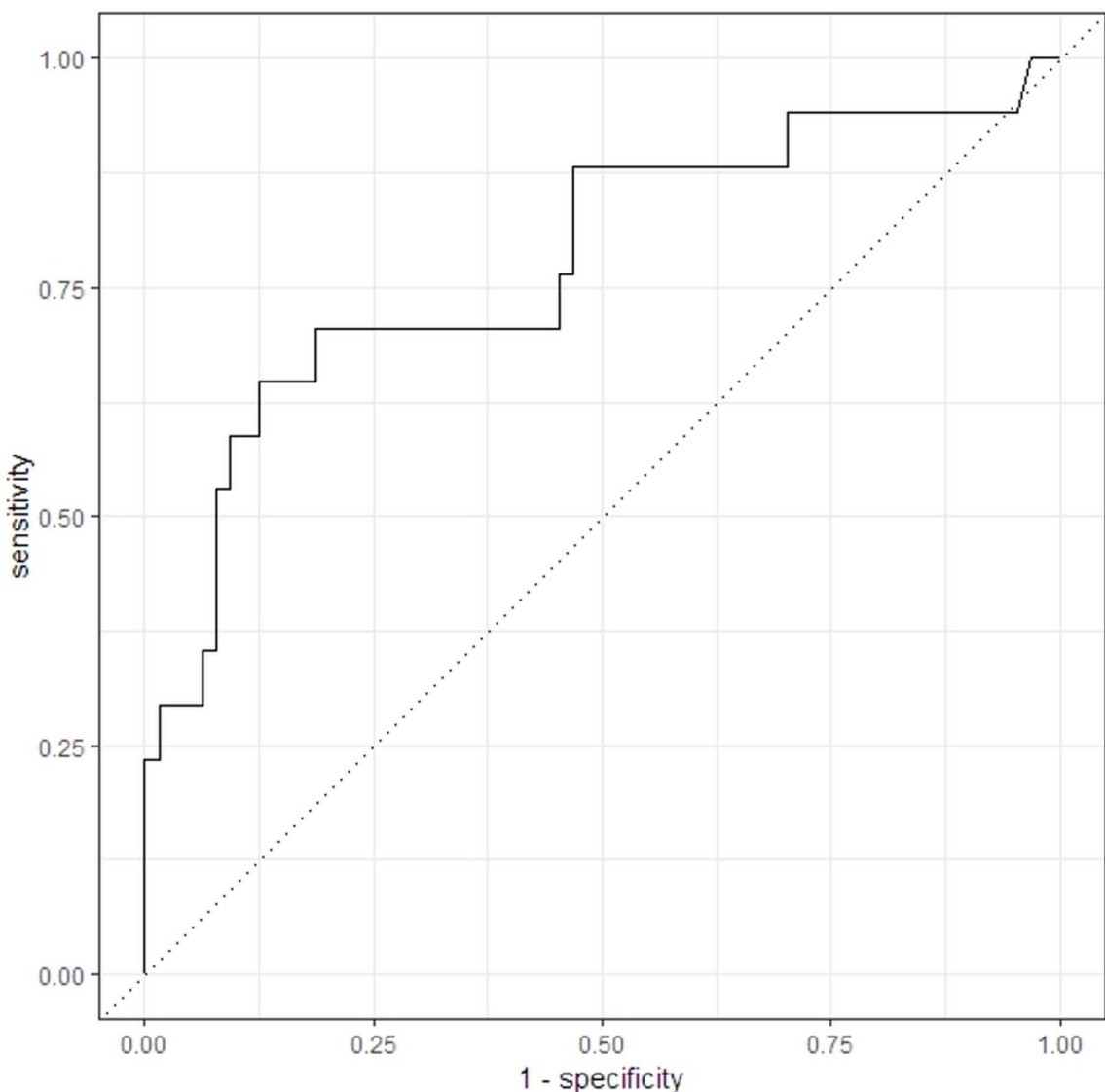

**Fig 4. Receiver operating characteristic curve of the RF model.**

limited sensitivity of 0.65. Overall, the screening performance of our classifier is considered acceptable [62,63]. This performance surpasses previous studies that utilized ML algorithms to study cognitive function in a Thai population. For example, a study using cardiovascular risk factors to compute the probability of mild cognitive impairment (MCI) in the Thai population reported AUCs ranging from 0.58 to 0.61 [64]. Another study applied phonemic verbal fluency (PVF) tasks combined with ML techniques to predict MCI in Thai participants, achieving an AUC of 0.73 [65]. Additionally, research on the Thai version of the CERAD neuropsychological battery for MCI screening reported the best model performance with an AUC of 0.77 [66]. Our integration of serum proteomic data provided a deeper insight into the value of biological markers for enhancing model precision for cognitive outcomes in the Thai population. However, a recent study leveraging high-dimensional neuroimaging data to classify cognitive performance in Portuguese individuals achieved an accuracy of 86.67% [67]. Their higher accuracy compared to our RF model may be attributed to the

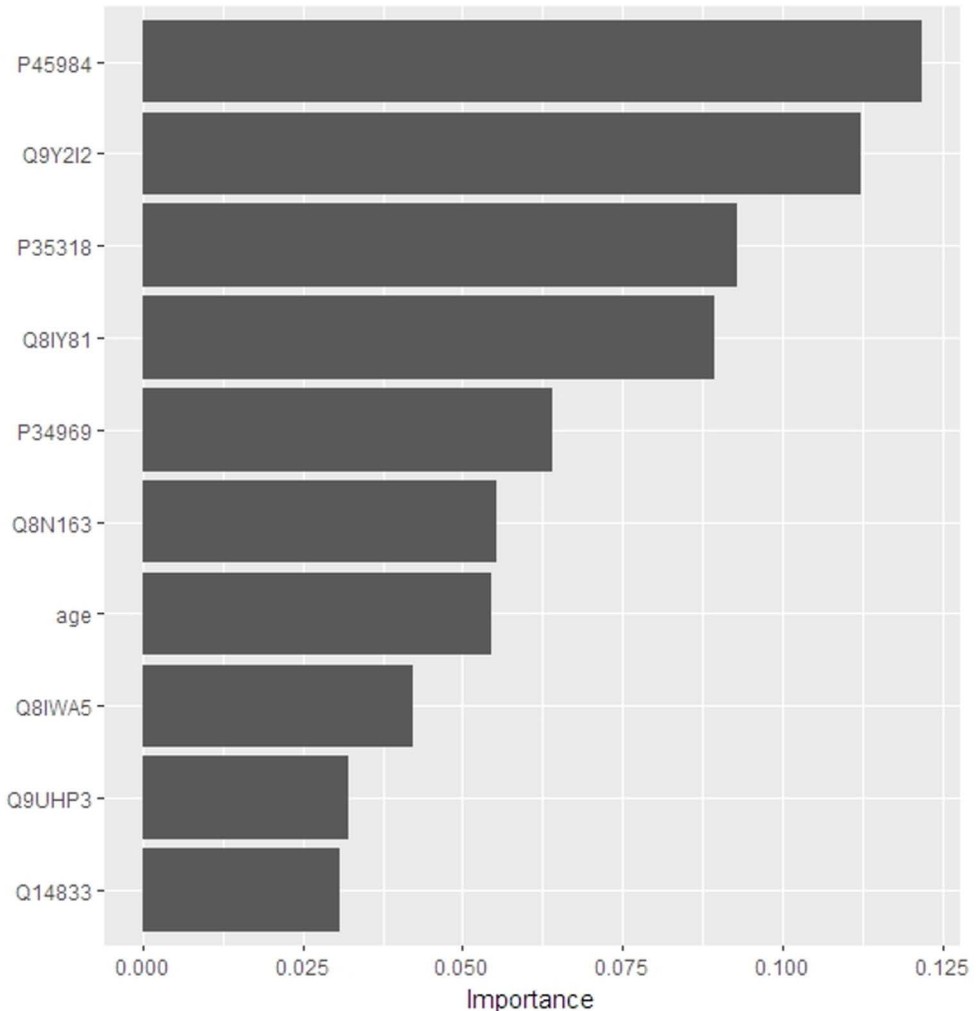

**Fig 5. Significance of variables in the RF model.**

direct relevance of neuroimaging features to cognition, while proteomic data reflects broader systemic processes that may introduce additional variability. Nevertheless, due to its limited sensitivity and moderate $F_1$-score [68], the performance of the RF model should be interpreted with caution. The low sensitivity was anticipated owing to the dataset's skew toward the negative cases (higher cognitive ability) [69,70] and the moderate sample size [71,72].

MAPK9 had the most impact on the RF model; it was enriched in the IL-17 signaling pathway along with another key variable, UPS25, with both DEPs being upregulated in the lower cognitive ability group. IL-17 signaling regulates inflammation by modulating inflammatory gene expression [73]. These proinflammatory factors, if unrestrained, may contribute to the pathology of a variety of autoimmune and chronic inflammatory conditions [74]. Dysfunctional and persistent inflammatory processes have been suggested as potential causative drivers of impaired cognitive functioning [75,76]. This is consistent with previous findings that proinflammatory molecules can cause the progression of brain deficits [77], and IL-17 reportedly initiates the onset of synaptic and cognitive impairments in the early stages of Alzheimer's disease [78], a process that may be mediated by activation of IL-17 receptors and MAPK [79].

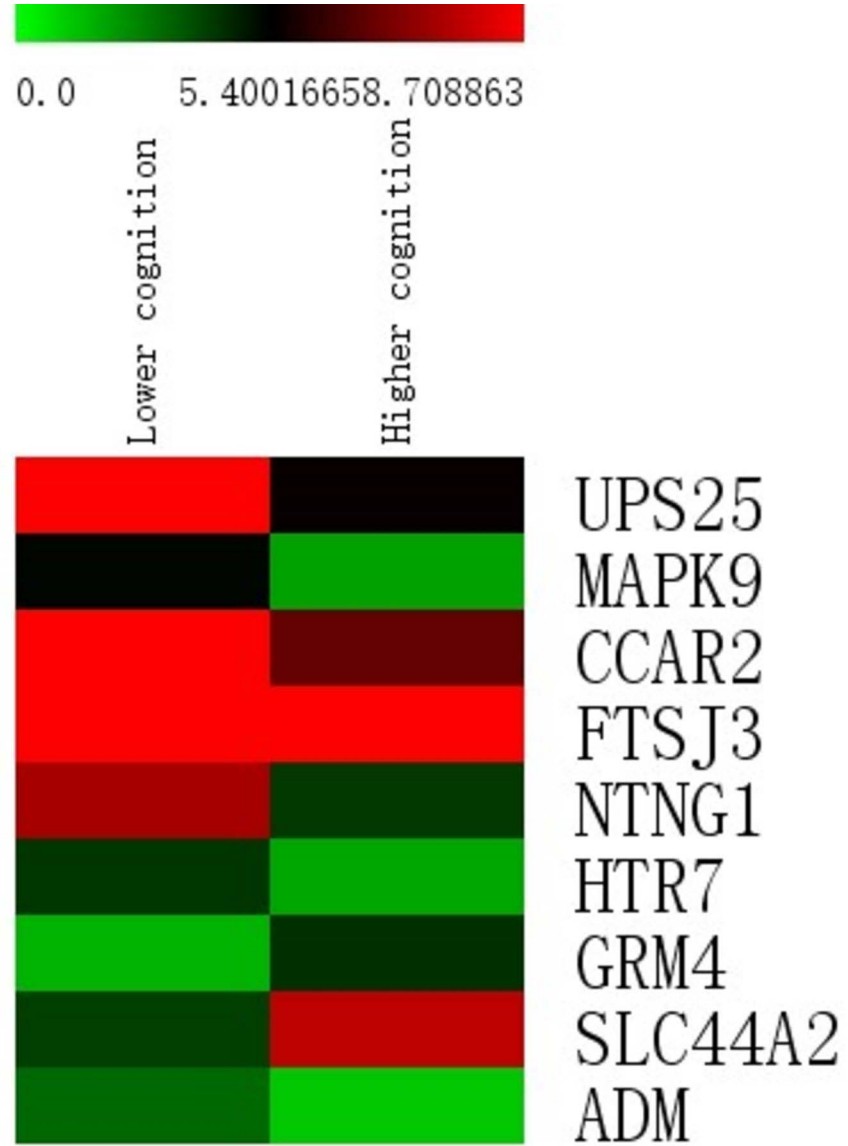

**Fig 6. Heat map of the top 9 protein variables.**

Furthermore, serotonin receptor 7 (HTR7), metabotropic glutamate receptor-4 (GRM4), and choline transporter-like protein 2 (SLC44A2) were also among the most influential variables. It has been demonstrated that inhibiting HTR7 modulates immune responses and decreases the severity of intestinal inflammation [80]. Serotonin stimulation of HTR7 activates downstream signaling modules such as MEK/MAPK [81]. This suggested that HTR7 may be positively connected with IL-17-mediated neuroinflammation and the poor cognitive functioning resulting from brain inflammation. This notion is supported in part by a prior result that HTR7 antagonism may have positive effects on schizophrenia-like cognitive deficits [82].

According to previous animal research, GRM4 controls adaptive immunity and restrains neuroinflammation [83]. Another study found that mutant mice lacking GRM4

were less capable of learning and integrating new spatial information into previously generated memory traces [84]. The involvement of spatial information learning in the WCST process has been proposed [85]. Choline transporter-like protein 2 (SLC44A2) is a high-affinity choline carrier [86]. Choline is an indispensable constituent in the biosynthesis of acetylcholine (ACh), and its transportation into the presynaptic terminals of cholinergic neurons requires a high-affinity choline transporter [87]. ACh is assumed to play an essential role in executive function, and cholinergic decline is related to poor WCST performance in healthy individuals [88]. Furthermore, SLC44A2 is a newly discovered plasma membrane and mitochondrial ethanolamine transporter [89], and ethanolamine has been linked to anti-inflammatory effects [90]. The downregulation of GRM4 and SLC44A2 in the lower cognitive ability group samples provides a potential biological mechanism for why subjects in this group performed poorly on the WCST. However, the weights of SLC44A2 and GRM4 are small, consistent with there being multiple interactions across diverse neurotransmitter systems in maintaining central nervous system homeostasis [91,92].

Netrin-G1 (NTNG1) and pro-adrenomedullin (ADM) are the second and third most important variables, and both DEPs were upregulated in the lower cognitive ability group samples. NTNG1 is a member of the Netrin family that is little studied in the brain but is implicated in inflammatory processes, including microglial function [93]. ADM has long been thought to be a biomarker for cognitive impairment [60,94]. ADM accumulation in the human brain contributes to memory loss with age [61], and blood ADM levels elevate in multiple pathological states, such as acute ischemic stroke and vascular cognitive decline with white matter alterations [95]. There is evidence indicating that inflammatory states stimulate the *in vivo* production of ADM [96]. These data, together with the evidence shown above, suggest that those in the lower cognitive ability group may have higher levels of neuroinflammation, resulting in poor cognitive performance on WCST. Age ranks as the seventh most important predictor in the model, with a moderate weight. Although it is widely accepted that aging is positively correlated to cognitive decline [44,97], our findings imply that this variable did not emerge as an influential factor in classifying cognitive variability among healthy Thai subjects.

This study has limitations that should be noted. First, the lower and higher cognitive ability groups were defined by their WCST % Errors > 1SD from the mean, but there is a controversy about this cutoff for defining an abnormal score on the WCST [34], and further study with different cutoff points might be needed to replicate the findings of the current study. Second, only age and proteomics data were included in the RF model. Given the potential direct and indirect interactions among multiple factors influencing cognitive function [98], additional factors such as socioeconomic status, lifestyle factors, and cardiovascular risk factors should be included in the model to improve model performance. Third, although the WCST has demonstrated validity and reliability as a stand-alone cognitive assessment tool for healthy individuals of various ages and educational backgrounds [99], and %Errors offers a general measure of performance on the WCST that closely correlates with FSIQ [31], variability in different domains of cognition requires a more comprehensive battery of cognitive testing. This would have been beyond the scope of the current study. Fourth, due to the moderate sample size and imbalanced number of subjects in the lower and higher cognitive ability groups, generalizing the current findings should be done with caution. Last, although the current statistical analysis outlines its benefits, it does not cover additional metrics such as effect size measures and confidence intervals. Including these metrics in future research will provide a more comprehensive evaluation and strengthen the robustness and interpretability of the findings.

## Conclusion

This study highlights the significant association between serum protein expression profiles and cognitive variability in a healthy Thai population using ML algorithms. By identifying 213 DEPs, with 155 upregulated in the lower cognition group and enriched in the IL-17 signaling pathway, our findings indicate a strong link between neuroinflammation and cognitive performance differences. The RF model using this proteomic data demonstrates strong classification accuracy and performance, with high specificity, outperforming previous studies that employed ML algorithms to examine cognitive function in Thai populations. This research underscores the critical role of biological markers in enhancing cognitive prediction and contributes to the development of more accurate ML models for diverse populations. These findings advance our understanding of cognitive variability and pave the way for early detection of cognitive disorders, offering promising directions for future research in cognitive health.

## Supporting information

**S1 Table. WCST scores and protein expression levels.**
(XLSX)

**S1 File. Author-generated code.**
(DOCX)

## Acknowledgments

The authors would like to thank all participants in this study. We would also like to thank the Faculty of Medical Science, Naresuan University and The National Center for Genetic Engineering and Biotechnology, Pathum Thani, Thailand for the facility supports.

## Author contributions

**Conceptualization:** Chen Chen, Samur Thanoi, Gavin P Reynolds, Sutisa Nudmamud-Thanoi.

**Data curation:** Chen Chen, Bupachad Khanthiyong, Benjamard Thaweetee-Sukjai, Sittiruk Roytrakul.

**Formal analysis:** Chen Chen, Sutisa Nudmamud-Thanoi.

**Funding acquisition:** Sutisa Nudmamud-Thanoi.

**Methodology:** Chen Chen, Sawanya Charoenlappanit, Sittiruk Roytrakul, Phrutthinun Surit, Ittipon Phoungpetchara.

**Project administration:** Sutisa Nudmamud-Thanoi.

**Supervision:** Samur Thanoi, Gavin P Reynolds, Sutisa Nudmamud-Thanoi.

**Writing – original draft:** Chen Chen.

**Writing – review & editing:** Samur Thanoi, Gavin P Reynolds, Sutisa Nudmamud-Thanoi.

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
