## [Decision Letter · Decision Letter 0]

29 Nov 2024

PONE-D-24-43711Proteomic associations with cognitive variability as measured by the Wisconsin Card Sorting Test in a healthy Thai population: A machine learning approachPLOS ONE

Dear Dr. Chen,

Thank you for submitting your manuscript to PLOS ONE. After careful consideration, we feel that it has merit but does not fully meet PLOS ONE’s publication criteria as it currently stands. Therefore, we invite you to submit a revised version of the manuscript that addresses the points raised during the review process.

We look forward to receiving your revised manuscript.

Kind regards,

Nafisa M. Jadavji, PhD, MSc, BSc

Academic Editor

PLOS ONE

**Journal Requirements:**

This work was partially supported by Global and Frontier Research University Fund, Naresuan University (grant number R2567C003) and Reinventing University Program 2024, the Ministry of Higher Education, Science, Research and Innovation (MHESI), Thailand (grant number R2567A141).

This work was partially supported by Global and Frontier Research University Fund, Naresuan University (grant number R2567C003) and Reinventing University Program 2024, the Ministry of Higher Education, Science, Research and Innovation (MHESI), Thailand (grant number R2567A141). The authors would like to thank all participants in this study. We would also like to thank the Faculty of Medical Science, Naresuan University and The National Center for Genetic Engineering and Biotechnology, Pathum Thani, Thailand for the facility supports.

This work was partially supported by Global and Frontier Research University Fund, Naresuan University (grant number R2567C003) and Reinventing University Program 2024, the Ministry of Higher Education, Science, Research and Innovation (MHESI), Thailand (grant number R2567A141).

Reviewers' comments:

Reviewer's Responses to Questions

**Comments to the Author**

1. Is the manuscript technically sound, and do the data support the conclusions?

Reviewer #1: No

Reviewer #2: No

Reviewer #3: Partly

2. Has the statistical analysis been performed appropriately and rigorously? 

Reviewer #1: No

Reviewer #2: No

Reviewer #3: Yes

3. Have the authors made all data underlying the findings in their manuscript fully available?

Reviewer #1: No

Reviewer #2: No

Reviewer #3: Yes

4. Is the manuscript presented in an intelligible fashion and written in standard English?

Reviewer #1: No

Reviewer #2: No

Reviewer #3: Yes

5. Review Comments to the Author

**Reviewer #1:**  Firstly, the format is not standardized and requires careful organization. Each section is not fulfilling its intended purpose. For example, the first section should analyze existing literature, identify innovative points, and describe the work of this paper.

Secondly, the abstract contains too much content. It should briefly explain the background, the work of this paper, and the contributions.

Thirdly, the level of innovation is insufficient.

Fourthly, there are no experimental charts or descriptions of the work conducted in this paper.

**Reviewer #2:**  Manuscript Number: PONE-D-24-43711

This work proposes " Proteomic associations with cognitive variability as measured by the Wisconsin Card Sorting Test in a healthy Thai population: A machine learning approach" to explore the association between serum protein expression profiles and one measure of cognitive variability. Having read this work, the following comments should be solved to enhance this work performance:

1-Please revise the abstract, discuss the problem statement and solution that this work presents, and discuss the outcome of this work in a better form than the current format. Furthermore, it's not well-preferred to include citations in the abstract; however, any references cited in the abstract must be given in full.

2-Please revise the introduction based on the following key points: background and motivation, objective, contributions, and the organization of this work. Please highlight the contributions of this work in the introduction. Furthermore, please provide a comprehensive literature review, the authors are advised to include this section under a Table format showing the year of publication, problem statements, techniques/ methods used to solve these problems, and advantages/disadvantages of each work. Please emphasize the benefits of this work over the mentioned works in the Table format.

3-Please add numbers to all sections and subsections.

4-Some of the presented sections like Bioinformatic analysis, Preprocessing, Machine learning model, Model validation, and Performance metrics are explained in a way that looks too general. Readers cannot understand these sections due to the information shortcomings. The authors are advised to enhance the readability of this work by presenting the entire design with the mathematical construction if available.

5-Please enhance the resolution quality of the presented figures.

6-Please include each of the presented figures in their exact locations directly.

7-To enhance the readability of this work, the authors are advised to include a flowchart or some graphical representation showing the overall working procedure based on the presented mathematical design.

8-One of the main drawbacks of this work is the statistical analysis, which does not clearly define its benefits. Please consider adding/discussing other statistical metrics.

9-Regarding the outcome of this work, the authors are advised to provide further comparisons with other works showing different results that have already been published by solid publishers to verify the effectiveness of this work clearly. Please show these differences in terms of graphical and statistical analysis.

10-Please revise the conclusion part. Currently, this section is unfit for publication. In the conclusion, the discussion of results includes several data points. It would be more effective to focus on the main findings, such as whether the proposed work outperforms others in the comparison.

11-Please clearly discuss the limitations of this work in the conclusion.

12-Please revise the English language and fix all the typos across the manuscript.

13-Regarding the reference list, please consider adding more recent references from the last 2-4 years and if available provide the DOI for each reference.

Finally, please solve all of the points mentioned above to enhance this work performance.

-End

**Reviewer #3:**  - Pay attention and rewrite the abstract in such a way as to exclude the references in the abstract. When reading the paper, I felt that the authors were just trying to cut and paste from the paper to create the abstract. Rather than that, try to summarize and make it more interesting.

- I suggest adding more information here: “Interindividual variability in cognitive trajectories has been observed in community-dwelling older adults across different cognitive domains, influenced by factors such as health status, education, and lifestyle.” In which cognitive domains? This provides clarity and specificity about the domains.

- I highly suggest rephrasing this for simplicity: "As a result, investigating the complex interactions between such factors using information technology and computer-based algorithms might enhance our understanding of the variability in normal cognition."

- The prior background (lines 66-71): the way the argument is presented lacks clarity and coherence, making it harder to follow the connection. I highly advise rewriting it in a way that the reader can get a clear idea without overloading them with jargon without context.

- It is unclear what role schooling plays. While it mentions that schooling attenuates cognitive differences, it doesn’t clearly explain the mechanism or the role it plays in the broader context of NMDAR or the cholinergic-estrogen interaction. Additionally, "schooling" is not a good term.

- Use the abbreviation "ML" for "machine learning" from the beginning. Also, it should be "ML algorithms"; the term "methods" (line 76) is not appropriate.

- Explain how racial bias is an issue for ML algorithms.

- Data cohort information is incomplete (e.g., sex).

- Why is the data split 60% to 40%? Any specification?

- Reading the paper, I felt it sounds robotic. So I highly suggest rewriting the methods section. For example, the authors mentioned that they used sampling and 60% of data for training, but they forgot to mention the number of subjects. This is quite hard to follow.

- There is no need to mention SMOTE too much, it's disrupting your original flow.

- The section "Machine Learning Model" must be rewritten. It includes too much unnecessary information rather than what the reader should focus on. For example, focus on the ML algorithm rather than the R packages.

- In the model validation section, rather than starting with "Cross-validation is a recommended ...," first introduce what cross-validation is and why it is important to your research.

- A proper introduction to performance metrics is needed.

- "Demographic data of the study population" should be moved to the dataset information.

- Justify why the authors used only the RF algorithm.

6. PLOS authors have the option to publish the peer review history of their article (what does this mean? ). If published, this will include your full peer review and any attached files.

**Do you want your identity to be public for this peer review?** For information about this choice, including consent withdrawal, please see our Privacy Policy .

Reviewer #1: No

Reviewer #2: No

Reviewer #3: No

---

## [Author Response · Author response to Decision Letter 0]

8 Jan 2025

Response to editor

1. The revised manuscript has been reformatted based on journal requirements.

2. Author-generated code has been uploaded as a supplementary file.

3. Our amended funding information has been included in the cover letter.

4. The funding information has been removed from the Acknowledgements section.

5. All figures have been removed from within the main text and uploaded as separate files.

Response to the reviewers

General comments:

In revising the manuscript in response to the reviewers’ comments, we have taken into account the requirements of the journal in the content and format of the individual sections. This includes an improved layout of the introduction with a brief review of the relevant background literature.

Reviewer #1: Firstly, the format is not standardized and requires careful organization. Each section is not fulfilling its intended purpose. For example, the first section should analyze existing literature, identify innovative points, and describe the work of this paper.

Author response: We appreciate the reviewer’s suggestion. Firstly, the manuscript has been reformatted entirely. The first section (Introduction) has been completely revised [Lines 44-87]. We conducted a literature review and outlined the background, objectives, innovative points, and contributions of this paper.

Secondly, the abstract contains too much content. It should briefly explain the background, the work of this paper, and the contributions.

Author response: We appreciate the reviewer’s suggestion. The abstract has been thoroughly revised to provide an explanation of the background, the work, and the contributions of this study [Lines 25-41].

Thirdly, the level of innovation is insufficient.

Author response: We appreciate the reviewer’s feedback and understand their concern regarding the level of innovation in our study. However, our research uniquely integrates proteomics data with cognitive assessments, utilizes machine learning techniques, and focuses on a specific ethnic cohort. These elements bring novel insights and contribute to an improved predictive model for normal cognitive variability. This is now explicitly described in the discussion [Lines 265-267].

Fourthly, there are no experimental charts or descriptions of the work conducted in this paper.

Author response: We appreciate the reviewer’s suggestion. We have included a detailed experimental chart to illustrate the overall workflow of our study [Fig. 1].

Reviewer #2: Manuscript Number: PONE-D-24-43711

This work proposes " Proteomic associations with cognitive variability as measured by the Wisconsin Card Sorting Test in a healthy Thai population: A machine learning approach" to explore the association between serum protein expression profiles and one measure of cognitive variability. Having read this work, the following comments should be solved to enhance this work performance:

1-Please revise the abstract, discuss the problem statement and solution that this work presents, and discuss the outcome of this work in a better form than the current format. Furthermore, it's not well-preferred to include citations in the abstract; however, any references cited in the abstract must be given in full.

Author response: We appreciate the reviewer’s suggestion. The abstract has been thoroughly revised to discuss the problem statement, the solution, and the outcomes of this study, following the required format of the journal [Lines 25-41]. We have now removed the citation in the abstract.

2-Please revise the introduction based on the following key points: background and motivation, objective, contributions, and the organization of this work. Please highlight the contributions of this work in the introduction. Furthermore, please provide a comprehensive literature review, the authors are advised to include this section under a Table format showing the year of publication, problem statements, techniques/ methods used to solve these problems, and advantages/disadvantages of each work. Please emphasize the benefits of this work over the mentioned works in the Table format.

Author response: We appreciate the reviewer’s suggestion. The Introduction section has undergone a thorough revision. This section presents the background and motivation, outlined the objectives of this paper, and highlighted the unique contributions of our study [Lines 44-87]. In this section, we have now provided a brief literature review as indicated in the journal requirements.

3-Please add numbers to all sections and subsections.

Author response: We appreciate the reviewer’s suggestion. After thoroughly reviewing the PLOS ONE submission guidelines and examining published papers in the journal, we have noted that section numbering is not a standard requirement. To align our manuscript with the journal’s established style and formatting guidelines, we have not added numbering to the sections and subsections.

4-Some of the presented sections like Bioinformatic analysis, Preprocessing, Machine learning model, Model validation, and Performance metrics are explained in a way that looks too general. Readers cannot understand these sections due to the information shortcomings. The authors are advised to enhance the readability of this work by presenting the entire design with the mathematical construction if available.

Author response: We appreciate the reviewer’s suggestion. We understand the importance of detailed explanations in enhancing the readability of our work. However, as we have employed pre-existing R packages and online analysis tools, detailed mathematical formulations would not be appropriate, following the convention with other publications in this field. Instead, we have revised the Bioinformatic analysis, Preprocessing, Machine learning model, Model validation, and Performance metrics sections to enhance clarity and ensure they are comprehensive and easily understandable [Lines 115-129, 132-147, 150-158,161-171, 174-183].

5-Please enhance the resolution quality of the presented figures.

Author response: The clarity and quality of the original figures should be no problem. However, we will improve their resolution clarity and update them in this revision.

6-Please include each of the presented figures in their exact locations directly.

Author response: We appreciate the reviewer’s suggestion. In accordance with the journal’s requirements, we have removed all figures from the main text and uploaded them as separate files.

7-To enhance the readability of this work, the authors are advised to include a flowchart or some graphical representation showing the overall working procedure based on the presented mathematical design.

Author response: We appreciate the reviewer’s suggestion. We have included a detailed flowchart to illustrate the overall workflow of our study [Fig. 1].

8-One of the main drawbacks of this work is the statistical analysis, which does not clearly define its benefits. Please consider adding/discussing other statistical metrics.

Author response: We appreciate the reviewer’s suggestion. We have updated the Methods section to provide a clear explanation of the benefits of the statistical analysis used in our study [Lines 117-122, 125-126, 163-164, 174-182]. However, there is a limitation of this study: although the current statistical analysis outlines its benefits, it does not provide additional metrics. Alternative analyses to complement the current approach would provide a more comprehensive evaluation and is an indication for future research. This has been included in the limitations of the study and is an indication for future research [Lines 346-349].

9-Regarding the outcome of this work, the authors are advised to provide further comparisons with other works showing different results that have already been published by solid publishers to verify the effectiveness of this work clearly. Please show these differences in terms of graphical and statistical analysis.

Author response: We have included a comparison with other studies that used ML algorithms to predict cognitive function in both Thai and Portuguese populations [Lines 257-272].

10-Please revise the conclusion part. Currently, this section is unfit for publication. In the conclusion, the discussion of results includes several data points. It would be more effective to focus on the main findings, such as whether the proposed work outperforms others in the comparison.

Author response: We have completely revised the conclusion to emphasize the main findings of this study and clarify how our work outperforms previous studies [Lines 351-363].

11-Please clearly discuss the limitations of this work in the conclusion.

Author response: The limitations of this study have been clearly discussed and are detailed in the final paragraph of the Discussion section [Lines 330-349].

12-Please revise the English language and fix all the typos across the manuscript.

Author response: We appreciate the reviewer’s suggestion. The entire text has been proofread by a native English-speaking co-author and all grammatical errors and typos have been corrected.

13-Regarding the reference list, please consider adding more recent references from the last 2-4 years and if available provide the DOI for each reference.

Author response: We appreciate the reviewer’s suggestion. We have included additional references from the last 2-4 years [ref 1, 8, 9, 13, 14, 25, 26, 52, 65, 67], and the DOI for each reference has been provided.

Finally, please solve all of the points mentioned above to enhance this work performance.

Author response: We have resolved all the points mentioned above.

-End

Reviewer #3: - Pay attention and rewrite the abstract in such a way as to exclude the references in the abstract. When reading the paper, I felt that the authors were just trying to cut and paste from the paper to create the abstract. Rather than that, try to summarize and make it more interesting.

Author response: We appreciate the reviewer’s suggestion. We have completely rewritten the abstract to effectively summarize this study. We have removed the citation in the abstract [Lines 25-41].

- I suggest adding more information here: “Interindividual variability in cognitive trajectories has been observed in community-dwelling older adults across different cognitive domains, influenced by factors such as health status, education, and lifestyle.” In which cognitive domains? This provides clarity and specificity about the domains.

Author response: The cognitive domains have been added, and the sentence has been rephrased as “variability in cognitive trajectories has been observed in community-dwelling older adults across different cognitive domains, such as episodic memory, vocabulary, executive function, attention, and psychomotor speed” [Lines 47-50].

- I highly suggest rephrasing this for simplicity: "As a result, investigating the complex interactions between such factors using information technology and computer-based algorithms might enhance our understanding of the variability in normal cognition."

Author response: This sentence has been rephrased to improve its readability as “Thus, using information technology and computer-based algorithms to probe these various complex interconnections offers great potential for enhancing our understanding of cognitive variability” [Lines 68-70].

- The prior background (lines 66-71): the way the argument is presented lacks clarity and coherence, making it harder to follow the connection. I highly advise rewriting it in a way that the reader can get a clear idea without overloading them with jargon without context.

Author response: The prior background section has been revised and supplemented with new evidence [ref 10] to strengthen the support for the paper and enhance readability for the audience [Lines 58-66].

- It is unclear what role schooling plays. While it mentions that schooling attenuates cognitive differences, it doesn’t clearly explain the mechanism or the role it plays in the broader context of NMDAR or the cholinergic-estrogen interaction. Additionally, "schooling" is not a good term.

Author response: Experiences and knowledge gained through education alter the cholinergic pathway activity, leading to an attenuation of these sex-dependent cognitive differences. We have rephrased it and explained the mechanism of education in attenuating cognitive sex differences. [Lines 64-66]. In addition, the word “schooling” has been substituted by “education”.

- Use the abbreviation "ML" for "machine learning" from the beginning. Also, it should be "ML algorithms"; the term "methods" (line 76) is not appropriate.

Author response: We have revised the main text and the abbreviation “ML” has been used to represent “machine learning” from the beginning. The term” machine learning methods” has been replaced with “ML algorithms” [Line 75].

- Explain how racial bias is an issue for ML algorithms.

Author response: When using ML algorithms, the racial and ethnic background of subjects is an essential issue to consider, since racial bias is a prevalent challenge facing ML in human studies, with consequences that can lead to racial disparities in healthcare access and outcomes [ref 23]. This has been revised in the text [Lines 74-78].

- Data cohort information is incomplete (e.g., sex).

Author response: In this study of 199 healthy subjects, 55.3% are female (n=110). When grouped, females represent 53.7% in the lower cognitive ability group (n=22), whereas they constitute 55.7% in the higher cognitive ability group (n=88). The details are outlined in the section “Demographic data of the study population” [Line 187] and in Table 1 [Line 195].

- Why is the data split 60% to 40%? Any specification?

Author response: The training and testing datasets were respectively proportioned at 0.6 (n=119) and 0.4 (n=80) of the total sample to optimize the balance between training and validation sets. This approach is designed to enhance the model's generalization capability and reduce the risk of overfitting. By allocating a sufficient portion of the data to the training set while preserving a substantial validation set, we can achieve more reliable and robust model performance, consistent with best practices in ML [ref 41]. This has also been added to the methods section [Lines 132-137].

- Reading the paper, I felt it sounds robotic. So I highly suggest rewriting the methods section. For example, the authors mentioned that they used sampling and 60% of data for training, but they forgot to mention the number of subjects. This is quite hard to follow.

Author response: We appreciate the reviewer’s suggestion. The numbers of subjects in both training and testing datasets have been added [Lines 132-133].

- There is no need to mention SMOTE too much, it's disrupting your original flow.

Author response: We appreciate the reviewer’s suggestion. The description of SMOTE has been revised and shortened [Lines 138-142].

- The section "Machine Learning Model" must be rewritten. It includes too much unnecessary information rather than what the reader should focus on. For example, focus on the ML algorithm rather than the R packages.

Author response: The Machine learning model section has been revised thoroughly, added details for the ML algorithm, and deleted unnecessary information [Lines 150-158].

- In the model validation section, rather than starting with "Cross-validation is a recommended ...," first introduce what cross-validation is and why it is important to your research.

Author response: We appreciate the reviewer’s suggestion. The meaning and importance of cross-validation have been added in the Model validation section [Lines 161-164].

- A proper introduction to performance metrics is needed.

Author response: We appreciate the reviewer’s suggestion. The introduction of all performance metrics has been added in the Performance metrics section [Lines 174-182].

- "Demographic data of the study population" should be moved to the dataset information.

Author response: We appreciate the reviewer’s suggestion. We understand the suggestion to move the demographic data to the Methods section. However, we believe that including the demographic

---

## [Decision Letter · Decision Letter 1]

22 Jan 2025

Proteomic associations with cognitive variability as measured by the Wisconsin Card Sorting Test in a healthy Thai population: A machine learning approach

PONE-D-24-43711R1

Dear Dr. Chen,

We’re pleased to inform you that your manuscript has been judged scientifically suitable for publication and will be formally accepted for publication once it meets all outstanding technical requirements.

Kind regards,

Nafisa M. Jadavji, PhD, MSc, BSc

Academic Editor

PLOS ONE

Additional Editor Comments (optional):

Dear Authors,

Thank-you for taking time to address the reviewers comments.

Sincerely,

Nafisa

Reviewers' comments:

Reviewer's Responses to Questions

**Comments to the Author**

1. If the authors have adequately addressed your comments raised in a previous round of review and you feel that this manuscript is now acceptable for publication, you may indicate that here to bypass the “Comments to the Author” section, enter your conflict of interest statement in the “Confidential to Editor” section, and submit your "Accept" recommendation.

Reviewer #2: All comments have been addressed

Reviewer #3: All comments have been addressed

2. Is the manuscript technically sound, and do the data support the conclusions?

Reviewer #2: Yes

Reviewer #3: Yes

3. Has the statistical analysis been performed appropriately and rigorously? 

Reviewer #2: Yes

Reviewer #3: Yes

4. Have the authors made all data underlying the findings in their manuscript fully available?

Reviewer #2: Yes

Reviewer #3: Yes

5. Is the manuscript presented in an intelligible fashion and written in standard English?

Reviewer #2: Yes

Reviewer #3: Yes

6. Review Comments to the Author

Reviewer #2: The authors successfully answered the reviewer's questions, so I recommend accepting this work for publication.

Reviewer #3: (No Response)

7. PLOS authors have the option to publish the peer review history of their article (what does this mean? ). If published, this will include your full peer review and any attached files.

**Do you want your identity to be public for this peer review?** For information about this choice, including consent withdrawal, please see our Privacy Policy .

Reviewer #2: No

Reviewer #3: No

---

## [Editor Report · Acceptance letter]

PONE-D-24-43711R1

PLOS ONE

Dear Dr. Chen,

I'm pleased to inform you that your manuscript has been deemed suitable for publication in PLOS ONE. Congratulations! Your manuscript is now being handed over to our production team.

Kind regards,

on behalf of

Dr. Nafisa M. Jadavji

Academic Editor

PLOS ONE